

# Female harbor seal (*Phoca vitulina*) behavioral response to playbacks of underwater male acoustic advertisement displays

Leanna P. Matthews[1], Brittany Blades[2] and Susan E. Parks[1]

[1] Biology Department, Syracuse University, Syracuse, NY, United States of America
[2] Oregon Coast Aquarium, Newport, OR, United States of America

Corresponding author
Leanna P. Matthews,
lematthe@syr.edu

## ABSTRACT

During the breeding season, male harbor seals (*Phoca vitulina*) make underwater acoustic displays using vocalizations known as roars. These roars have been shown to function in territory establishment in some breeding areas and have been hypothesized to be important for female choice, but the function of these sounds remains unresolved. This study consisted of a series of playback experiments in which captive female harbor seals were exposed to recordings of male roars to determine if females respond to recordings of male vocalizations and whether or not they respond differently to roars from categories with different acoustic characteristics. The categories included roars with characteristics of dominant males (longest duration, lowest frequency), subordinate males (shortest duration, highest frequency), combinations of call parameters from dominant and subordinate males (long duration, high frequency and short duration, low frequency), and control playbacks of water noise and water noise with tonal signals in the same frequency range as male signals. Results indicate that overall females have a significantly higher level of response to playbacks that imitate male vocalizations when compared to control playbacks of water noise. Specifically, there was a higher level of response to playbacks representing dominant male vocalization when compared to the control playbacks. For most individuals, there was a greater response to playbacks representing dominant male vocalizations compared to playbacks representing subordinate male vocalizations; however, there was no statistical difference between those two playback types. Additionally, there was no difference between the playbacks of call parameter combinations and the controls. Investigating female preference for male harbor seal vocalizations is a critical step in understanding the harbor seal mating system and further studies expanding on this captive study will help shed light on this important issue.

## INTRODUCTION

Acoustic communication is a critical component for reproductive success in a wide range of species: males can use reproductive signals to attract females or defend territories against other males (for reviews, see *Searcy & Andersson, 1986*; *Andersson, 1994*). Acoustic

reproductive displays have several advantages over visual signals in low visibility habitats, such as areas of dense foliage, underwater, or under low light conditions at night. For instance, signalers can use acoustic signals to rapidly transmit a large amount of information to potential receivers, and the animals do not have to see each other in order to communicate (for review, see *Endler, 1993*).

Female preference for acoustic signals plays an important role in sexual selection (*Andersson, 1994*). Females of some species have been shown to prefer signal traits that reflect honest indicators of male size, dominance status, or energetic reserves. For example, female red deer (*Cervus elaphus*) prefer calls with a lower frequency, which correspond to males of larger size (*Charlton, Reby & McComb, 2007*). Female tungara frogs (*Physalaemus pustulosus*) also prefer calls with a lower frequency, also indicating a preference for larger males (*Ryan, 1980*). Female Hermann's tortoises (*Testudo hermanni*) prefer fast-rate acoustic displays, which are related to hematocrit levels in males and represent high quality mates (*Galeotti et al., 2005*).

Females have also been shown to prefer males who demonstrate a higher rate of signal output. In red deer, females show preference for higher calling rates, which possibly correspond to male quality (*McComb, 1991*). A preference for high calling rates has also been seen in Gulf Coast toads (*Bufo valliceps*) (*Wagne Jr & Sullivan, 1995*). Likewise, female grey mouse lemurs (*Microcebus murinus*) prefer higher calling activity, which corresponds to the relative dominance status of the male (*Craul, Zimmermann & Radespiel, 2004*). Females of some species, such as the gray tree frog (*Hyla versicolor*) (*Gerhardt et al., 2000*), show preference for calls that have a longer duration, which is a reliable indicator of energetic expenditure.

The physics of the underwater environment, specifically the incompressible nature of water compared to air, allows for much more efficient sound propagation than in air. Sound travels approximately 340 m/s in air, while it can travel around 1,500 m/s in seawater, depending on temperature and salinity. Additionally, visual communication is difficult in the underwater environment, as it is light-limited and individuals need to maintain a line of sight for effective signaling. Therefore, species that spend all or most of their lives underwater, such as marine mammals, have evolved to rely on acoustic communication for many behaviors, including reproductive advertisement displays.

Harbor seals are a commonly occurring marine mammal species from the pinniped group that breed underwater. Previous studies have shown that select males establish and hold territories during the breeding season (*Van Parijs et al., 1997*; *Van Parijs, Janik & Thompson, 2000*; *Hayes et al., 2004a*). However, not all males hold territories and it does not necessarily correspond to higher reproductive success for individual male harbor seals (*Coltman, Bowen & Wright, 1998*; *Coltman, Bowen & Wright, 1999*; *Hayes et al., 2006*). During the breeding season, harbor seal males also produce underwater acoustic cues, known as roars. These signals are low in frequency (78–1,300 Hz) and can be up to 10 s in duration (*Hanggi & Schusterman, 1994*; *Van Parijs, Hastie & Thompson, 2000*; *Matthews et al., 2017*). *Nicholson (2000)* studied male-male interactions of harbor seals in Monterey Bay, CA from various age groups and hypothesized that subordinate males, males vocalizing alone, have roars that are shorter in duration and higher in frequency, while dominant

males, males that are attended by other males, have roars with longer duration and lower frequency. This prediction is consistent with studies of other mammalian species with older, dominant males producing longer, and lower frequency signals (e.g., *Vannoni & McElligott, 2008*; *Wyman et al., 2012*). There were also less frequent observations of intermediate vocalizations, i.e., high frequency, long duration, by males that were only occasionally attended (*Nicholson, 2000*). A recent publication on harbor seal vocalizations from a population in British Columbia, Canada indicates that male vocalizations form a continuum, with a distribution of both duration and frequency of male vocalizations (*Nikolich, Frouin-Mouy & Acevedo-Gutierrez, 2016*). This aligns with the *Nicholson (2000)* study, which indicated a range of characteristics for roar vocalizations, with hypothesized 'dominant' and 'subordinate' males representing the extremes of this continuum of vocal parameters.

A previous study demonstrated that roars were important for underwater territory establishment by males (*Hayes et al., 2004b*). In this study, wild male harbor seals in Elkhorn Slough, CA were exposed to three acoustic stimuli based on the results of *Nicholson (2000)*: long duration and low frequency roars (dominant male signals), short duration and high frequency roars (subordinate male signals), and ambient water noise (control) (*Hayes et al., 2004b*). Male harbor seals responded most aggressively to stimuli representing signals produced by subordinate male roar vocalizations (*Hayes et al., 2004b*). There were no observed responses by females to any of the stimuli presented in the study, however it is hypothesized that the roars may also influence female preference for potential mates (*Hayes et al., 2004b*). It is possible that females are, in fact, responding to these male calls and have a preference for specific characteristics of acoustic signals, but this behavior has yet to be observed. The studies by *Nicholson (2000)* and *Hayes et al. (2004b)* served as the basis for the playback experiments described here.

A common approach for studying female response and preference for acoustic cues is via playbacks (e.g.: *Ryan, 1980*; *Hedrick, 1986*; *Searcy & Andersson, 1986*; *Catchpole, 1987*; *McComb, 1991*). Previous playback experiments have investigated call function in a variety of marine mammal species. In the first playback to marine mammals in the field, *Watkins & Schevill (1968)* played back recorded Weddell seal (*Letonychotes weddellii*) calls to male Weddell seals in order to test the call function of various vocalizations. They found that high quality recordings of conspecifics caused the subjects to respond acoustically, but they also noted that the recordings did not attract seals and that subsequent playbacks elicited less of a response (*Watkins & Schevill, 1968*). Other playbacks to cetaceans, including humpback whales (*Megaptera novaeangliae*) (*Tyack, 1983*; *Mobley Jr, Herman & Frankel, 1988*), southern right whales (*Eubalaena australis*) (*Clark & Clark, 1980*), and North Atlantic right whales (*Eubalaena glacialis*) (*Parks, 2003*), have tested reactions to conspecific calls and artificial calls. *Tyack (1983)* found that singing humpback whales ceased vocal activity and moved away in response to playbacks of humpback whale social sounds, and *Mobley Jr, Herman & Frankel (1988)* found that males approached the playback speaker when sounds of sexually mature females were played. In right whales, playbacks have indicated that southern right whales are able to differentiate between calls of conspecifics and humpback whales (*Clark & Clark, 1980*) and North Atlantic right whales respond to

playbacks of calls produced during right whale social interactions (*Parks, 2003*). There have also been additional playbacks to various pinnipeds that have investigated caller recognition: subantarctic fur seal (*Arctocephalus tropicalis*) females are able to recognize the vocalizations of pups (*Charrier, Mathevon & Jouventin, 2002*), Australian sea lion (*Neophoca cinerea*) pups can recognize the vocalizations of their mothers (*Charrier, Pitcher & Harcourt, 2009*), and northern fur seal (*Callorhinus ursinus*) mothers and pups can recognize vocalizations of each other, although there is higher energy expenditure on the part of the pups (*Insley, 2001*). These studies indicate that playbacks are a powerful tool for studying marine mammal behavior. Studying female responses to male acoustic signals in captivity is an excellent way to add further insight into the underwater behavior of these marine mammal species.

This study used playback experiments with captive individuals to investigate female response to male breeding vocalizations in harbor seals. The playbacks tested whether females approach male calls more than control signals that do not contain biologically significant signals. Multiple combinations of male call parameters (duration and frequency) were used to determine whether these parameters affected female preference. Based on playbacks to other marine mammal species, we predict that female harbor seals will have a greater response to playbacks with conspecific calls compared to a control. Additionally, based on previous work that indicates that females prefer lower frequency and longer duration calls, it is likely that female harbor seals will approach the playback speaker more often during playbacks of dominant male harbor seal calls, which have a lower frequency and longer duration.

## METHODS

Playback experiments were conducted at the Oregon Coast Aquarium in Newport, OR in the summer of 2015 and 2016 (Syracuse University IACUC Permit #14-003). Five female harbor seals were tested in the first year of trials and a subset of four female harbor seals were tested again for the second year. The fifth individual was not tested in the second year because she went into the molt early and would not enter the pool voluntarily. The individuals were all reproductively mature, demonstrating signs of estrous in previous years, and ranged in age from six to 30 years (Table 1). Four of the five individuals were born in the wild and stranded as pups. All were housed with male harbor seals when not isolated for the playback trials. Experiments were conducted while the individuals were in estrous, with the exception of three individuals during the 2015 season, who were tested after the molting period, which occurs after estrous. Estrous was determined using visual cues, including the appearance of inflamed genitalia.

Recordings of male harbor seal roars were collected in Elkhorn Slough, CA during May 2015. Recordings were made in close proximity to a single territorial individual to obtain a series of high quality calls. Close proximity was assumed given that a subset of 20 calls had a signal-to-noise ratio >10 dB. The recordings were divided into segments that each contained five roars. The duration and frequency of the roars were adjusted in Adobe Audition to create signals for the playback experiments. A total of 200 roars were modified as test signals.
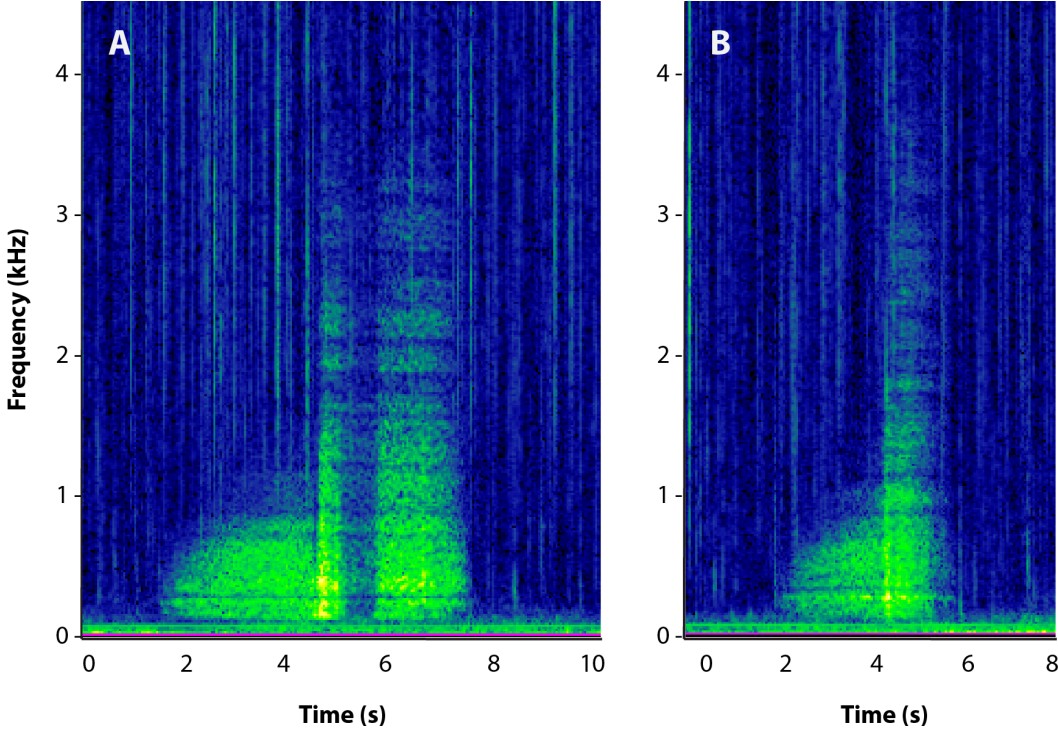

**Figure 1  Spectrograms of male roar vocalizations.** Visual representations of male harbor seal vocalizations that have been modified to represent dominant males (A) and subordinate males (B) (spectrogram parameters: Hann window, 50% overlap, discrete Fourier transform [DFT] size = 4,096).

**Table 1  List of Individuals from playback experiments.** Individuals used in playback experiments, their approximate dates of birth, life history information, and whether or not they were in estrous during the playbacks for either year of experiments. Boots was not included in the 2016 experiments.

| Individual | DOB | Life History | Estrous 2015 | Estrous 2016 |
|---|---|---|---|---|
| Boots | January 1988 | Wild born (California) | No | – |
| Pinky | June 1986 | Wild born (Washington) | No | Yes |
| Swap | August 1991 | Wild born (Washington) | Yes | Yes |
| Tater | March 1994 | Captive born (California) | No | Yes |
| Tazzy | 2009 | Wild born (British Columbia) | Yes | Yes |

Signals were created to test combinations of male call parameters. Two groups of playbacks represented the extremes of calls found in nature (Fig. 1): one for subordinate male roars (short duration, high frequency) (SH), and the other for dominant male roars (long duration, low frequency) (LL). These signals mimicked those used in the playback experiments of *Hayes et al. (2004b)*. The "subordinate" playbacks consisted of roars that were less than 2.5 s (mean ± SD: 2.10 ± 0.05 s) and had a minimum frequency of greater than 220 Hz (mean ± SD: 247.30 ± 8.80 Hz) (*Nicholson, 2000*; *Hayes et al., 2004b*). The "dominant" roars were greater than 3.0 s in length (mean ± SD: 3.33 ± 0.09 s) and a minimum frequency of less than 200 Hz (mean ± SD: 178.19 ± 8.63 Hz) (*Nicholson, 2000*;
*Hayes et al., 2004b*). The other two groups of playbacks represented combinations of the extremes of call parameters found in subordinate and dominant roars—short duration, low frequency (SL) (mean ± SD: 2.14 ± 0.66 s and 156.27 ± 45.45 Hz) and long duration, high frequency (LH) (mean ± SD: 3.54 ± 1.08 s and 264.70 ± 62.61 Hz). These call parameter combinations are less common in nature (*Nicholson, 2000*) and these playbacks were used to tease out the acoustic characteristics to which females respond most strongly.

A total of five playback files were made for each group of playback signals. Each file consisted of 1–2 min of active playback that contained five roars based on the natural roar timing of the recorded individual, and 1 min of silence, when no roars were present. The call intervals during the active playback varied slightly between files, with an average interval of 12.15 ± 8.95 s (mean ± SD). This series of active playback and silence was repeated eight times for a total of approximately 20 min.

Two additional groups of playback files were created as controls. The first control (W) consisted of only water noise recorded in Elkhorn Slough and contained no biologically significant sounds related to mating and territoriality. The secondary control (WT) was added as an additional control in the second year of experiments and consisted of water noise with an occasional synthetic tonal sound that was similar in frequency and duration to harbor seal roars. This tonal sound contained no biologically significant information and was used to ensure that female harbor seals were responding to the roar vocalizations in the experimental trials, rather than the occurrence of novel sounds.

Each individual was exposed to one playback file per day for three to four consecutive days, depending on the year. Three playbacks stimuli were used in the first year of experiments (LL, SH, and W) and four stimuli were used during the second year (SL, LH, W, and WT). Playbacks were arranged in a randomized block design, with three or four treatments (i.e., playbacks) and four or five blocks (i.e., individuals), depending on the year, and each subject was exposed to a unique series of playbacks to avoid pseudoreplication. All trials were conducted in the same enclosure with only one individual tested at a time. The individual was allowed in to the testing pool prior to the start of the playback and playbacks began after the individual had been swimming for at least three minutes and aquarium personnel had left the enclosure. Hunger state was not controlled for and no food rewards were given during the trials.

An underwater speaker (Lubell Labs LL916, frequency response: 200 Hz–23 kHz (±10 dB at 500 Hz–21 kHz)) was lowered approximately 1.5 m into the pool directly next to the wall. The speaker was housed in a PVC cage and was positioned in the same place for every trial. The speaker was connected to an amplifier (Dual XPA2100, frequency response: ± 3 dB at 20 Hz–20 kHz), and the amplifier was connected to an iPod, which was pre-loaded with the playback files. Received levels were measured throughout the pool at multiple depths to ensure that the playbacks were approximately equal in loudness to estimates of vocalization source levels (155 dB re 1 μPa: C Reichmuth, pers. comm., 2015). Measured levels of the roars in the playbacks ranged from 149 to 156 dB re 1 μPa rms. Since the playback experiments, more recent studies have shown that the average loudness of harbor seal vocalizations is 144 to 145 dB re 1 μPa rms and range from 129 to 149 dB re 1 μPa rms (*Casey, Sills & Reichmuth, 2016*; *Matthews et al., 2017*). A GoPro camera (HERO4 Silver)

was used to record all of the trials. The camera was placed on an overlook above the enclosure (approximately 5 m elevation), which allowed for full coverage of the playback pool, and the location of the camera was the same for all the trials.

Video footage was used to complete behavioral sequencing for each of the playback experiments. The numbers of approaches to the playback speaker were counted as a proxy for female response. Approaches were defined as a deliberate investigation of the speaker. This included any touching of the speaker with the vibrissae or curious examination of the PVC apparatus. The amount of time spent at the speaker for each approach was also measured. If a female approached the speaker, left, and approached again quickly, it was considered two separate approaches.

A series of Kruskal–Wallis tests were used to compare the number of approaches between playbacks with signals and control playbacks to determine if females were attracted to male calls in general. Three separate Kruskal–Wallis tests were run: one for a combined data set for both years of playback experiments, one for the first year of experiments, and one for the second year of experiments. Then, the numbers of approaches to the playback speaker for the different categories (first year of playbacks: LL, SH, W; second year of playbacks: SL, LH, W, WT) were compared using a nonparametric Friedman's test to test for preference between the playback types and account for differences between individuals. Post hoc comparisons using the Wilcoxon-Nemenyi-McDonald-Thompson test were used to further investigate significant comparisons (*Hollander, Wolfe & Chicken, 2013*). The two years of playbacks were analyzed separately because only a subset of the individuals was tested in the second year. A second set of Friedman's tests was used to compare the amount of time spent at the speaker. Statistical analyses were done in R version 3.2.3 (*R Core Team, 2013*).

## RESULTS

There was a significant difference between the numbers of approaches to the playback speaker when projecting male vocalizations as compared to the controls for the combined data of both years ($p = 0.003$ at $\alpha < 0.05$). When analyzing the years individually, the difference between the number of approaches for playbacks with signals and playbacks with controls was significant for the first year of experiments ($p = 0.009$, LL and SH vs. W), but not significant for the second year of experiments ($p = 0.27$, SL and LH vs. W and WT).

Friedman's tests indicated an overall significant difference in the number of approaches made to the different categories of playbacks for the first year of experiments (Fig. 2A, $p = 0.029$). Female harbor seals approached the playback speaker significantly more during the LL playback when compared to the control (W) ($p = 0.021$). The maximum number of approaches during LL male playbacks was 4, which was noted for two of the five individuals, and the minimum number was one approach. No individuals approached the speaker during the control (W) playback. There was no statistical difference between the number of approaches during the SH and LL male playbacks ($p = 0.377$) or the SH and control (W) playbacks ($p = 0.367$). For the second year of playbacks, there was no difference in the number of approaches to the speaker for any of the stimuli (SL, LH, W, and WT), with overall low numbers of approaches to all stimuli, including controls (Fig. 2B,

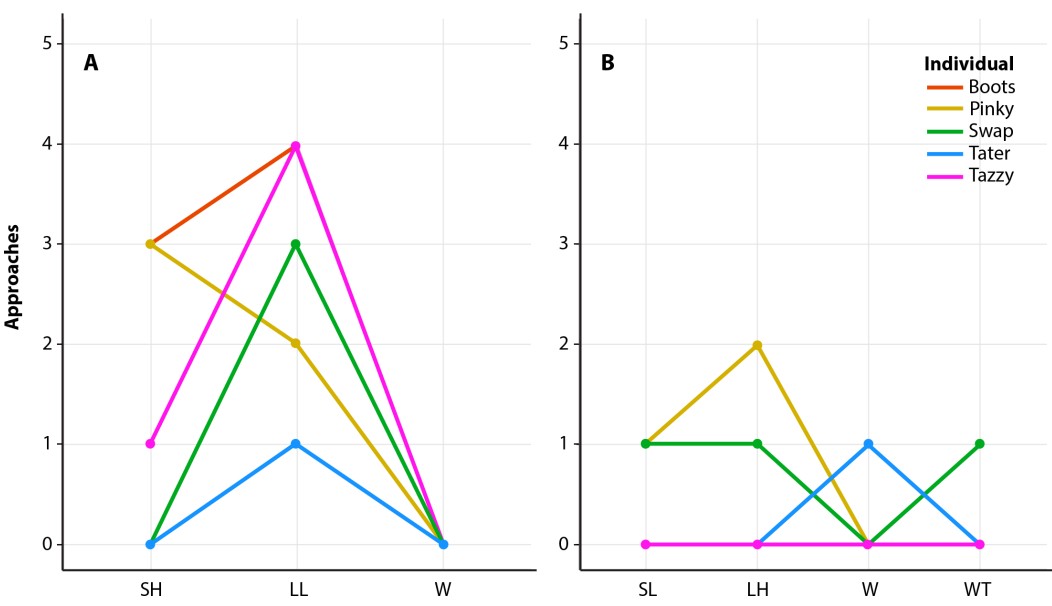

**Figure 2  Results from playback experiments.** Line graphs illustrating the number of approaches made to the playback speaker during the first year of playbacks (A) and second year of playbacks (B). Year one tested short duration, high frequency signals (SH), long duration, low frequency signals (LL), and a water noise control (W). Year two tested short duration, low frequency signals (SL), long duration, high frequency signals (LH), a water noise control (W), and a water noise control with tonal signals in the frequency range of male signals (WT).

$p = 0.733$). The maximum number of approaches was during the LH playback, with two approaches. One individual approached the speaker once during this playback and the other two individuals did not approach the speaker at all. Two individuals approached the SL playback once, while the other two did not approach the speaker. Only one individual approached the speaker during the W playback and one individual approached during the WT playback, though the individuals who approached were different. The other three females did not approach the speaker during either of the control playbacks (W or WT).

The results of the Friedman's tests for the amount of time spent at the speaker were similar to the approach results (year one: $p = 0.029$; year two: $p = 0.782$). In the first year of playbacks, approaches ranged from 1–13 total seconds, with the longest approach times occurring during LL male playbacks. In the second year, approaches lasted from 4–10 s. A summary of the time spent at the speaker during each approach is available in Table 2.

## DISCUSSION

This study is the first to specifically investigate if (1) female harbor seals show interest in male acoustic signals and (2) to test for potential female preference for different parameters of male breeding vocalizations in harbor seals. In regards to the first goal of this paper, the combined data from both years of playbacks, as well as only the data from the first year, indicate that captive female harbor seals show significantly more interest in playbacks with male vocalizations compared to controls. A previous study on male harbor seal response to

**Table 2  Summary statistics of the time spent investigating the speaker.** The average time spent investigating the speaker during each approach and the range of times observed for approaches for each trial during both years of playback experiments. The NA values in the range column correspond to trials for which there was only one approach.

| Year | | Dominant | | Subordinate | | Control 1 | |
|---|---|---|---|---|---|---|---|
| | Individual | Avg. (s) | Range (s) | Avg. (s) | Range (s) | Avg. (s) | Range (s) |
| 2015 | Boots[a] | 3.25 | 2–5 | 2.33 | 2–3 | 0.00 | 0.00 |
| | Pinky[a] | 3.50 | 1–6 | 3.67 | 2–6 | 0.00 | 0.00 |
| | Swap | 1.33 | 1–2 | 0.00 | 0.00 | 0.00 | 0.00 |
| | Tater[a] | 3.00 | NA | 0.00 | 0.00 | 0.00 | 0.00 |
| | Tazzy | 6.75 | 2–13 | 2.00 | NA | 0.00 | 0.00 |

| Year | | LF/short | | HF/long | | Control 1 | | Control 2 | |
|---|---|---|---|---|---|---|---|---|---|
| | Individual | Avg. (s) | Range (s) | Avg. (s) | Range (s) | Avg. (s) | Range (s) | Avg. (s) | Range (s) |
| 2016 | Pinky | 3.00 | NA | 3.50 | 3–4 | 0.00 | 0.00 | 0.00 | 0.00 |
| | Swap | 10.00 | NA | 4.00 | NA | 0.00 | 0.00 | 6.00 | NA |
| | Tater | 0.00 | 0.00 | 0.00 | 0.00 | 4. 00 | NA | 0.00 | 0.00 |
| | Tazzy | 0.00 | 0.00 | 0.00 | 0.00 | 0.00 | 0.00 | 0.00 | 0.00 |

Notes.
[a] Indicates individuals that were not in estrous during the playback experiments.

playbacks in the wild did not note any behavioral shifts in females (*Hayes et al., 2004b*), but the responses made by females in our experiments were fairly brief and were only detectable because there was a clear view to the bottom of the pool. It is possible that females did approach the playback speaker in the previous study, but the approaches were undetectable to the researchers due to turbidity of the water. There was no significant difference in response when comparing the call parameter combination playbacks of the second year (SL and LH) to the controls (W and WT). It is possible that both the lowest frequency and longest duration or the highest frequency and shortest duration is necessary for females to respond to vocalizations and that the other combinations of parameter extremes, which are less common in nature (*Nicholson, 2000*), are less likely to elicit a response.

The numbers of approaches to the playback speaker were not significantly different between the types of playbacks that contained male acoustic signals (year one: LL vs. SH, year two: SL vs. LH). However, although not statistically significant, four out of five females did approach the playback speaker more during playbacks of dominant (LL) vocalizations compared to playbacks of subordinate (SH) vocalizations. It is possible, but is not confirmed here, that the roar vocalization may play a role in male–female communication during the breeding season, with females using acoustic cues to make decisions on mate preference when other modalities, such as sight and smell, are limited. Females from other species have been shown to prefer vocalizations that are honest advertisements and denote a higher dominance rank (e.g., *Clutton-Brock & Albon, 1979*; *McComb, 1991*; *Craul, Zimmermann & Radespiel, 2004*; *Galeotti et al., 2005*; *Puechmaille et al., 2014*). It is also possible that the statistical insignificance between the LL and SH calls is representative of actual female responses, and there is no difference in response level between the two. This may indicate

that variability in roar vocalization parameters are primarily for the purpose of male-male interactions and not a mechanism of female preference.

Even though there was no statistically significant difference between any playback stimuli in the second year of trials, there were differences in responses of individuals to the different stimuli. For instance, Pinky responded more during the SL and LH trials compared to either of the controls (W and WT), while Swap responded equally to the SL, LH, and WT, but did not respond to the water noise control (W). Because of the variation in responses between individuals, it is possible that there may be other information in the roars that is important for female discrimination and preference.

It is also important to note that because different individuals were tested in the two years, it is not possible to statistically compare the responses to the LL male playbacks and the secondary control with the tonal signals (WT) using a Friedman's test. It would be of interest for future studies to further investigate these results, and to determine with certainty if females respond more to biologically important information or novel acoustic stimuli.

In the first year of trials, only two individuals were confirmed to be in estrous during the playback experiments. However, similar trends—an increase in the number of approaches during the LL male playback compared to the SH and control playback (W)—were observed between estrous and non-estrous individuals, with the exception of one individual. One female, Pinky, approached the speaker more during the SH male playback compared to the LL male playback. This could be due to a variety of factors. Firstly, she might have been unmotivated due to lack of estrous and her approaches were purely based on curiosity. Secondly, Pinky was the oldest of the test subjects and might not have as great an ability to discriminate acoustic signals as younger individuals due to potential presbycusis, although hearing data were not available for any of the individuals in this study.

Harbor seals inhabit a wide range and the vocal characteristics can vary between populations (*Van Parijs et al., 2003*). All females, despite their origin, were exposed to roars modified from an opportunistic data set of recordings made in California. It is possible that a different set of acoustic characteristics would yield different responses by females based on their genetic population of origin.

It is also possible that females were responding just to the presence of harbor seal vocalizations, and not specifically the roar; this preference between vocalization types has not been tested. However, the females in these trials did not investigate the playback speaker more than during the controls except during the dominant male playbacks, indicating that the novelty of a seal sound alone did not evoke a significantly stronger response.

## CONCLUSIONS

This study demonstrated that females appear to show interest in male calls. Future studies should aim to confirm if female harbor seals show similar interest in male vocalizations in the wild. The results also indicate preliminary evidence of a female preference for lower frequency, longer duration signals. A two-choice test for female harbor seals would help further parse out female responses to male acoustic signals.

## ACKNOWLEDGEMENTS

The authors would like to thank the staff and volunteers of the Oregon Coast Aquarium for their assistance with this project. We would also like to thank Alex Carbaugh-Rutland for his help in data collection during the 2016 season. IACUC approval for the playback experiments was provided by Syracuse University.

### Funding

This work was funded by the Marine Mammal Commission (MMC15-272) and a National Geographic Young Explorer's Grant. The funders had no role in study design, data collection and analysis, decision to publish, or preparation of the manuscript.

### Grant Disclosures

The following grant information was disclosed by the authors:
Marine Mammal Commission: MMC15-272.
National Geographic Young Explorer's Grant.

### Competing Interests

The authors declare there are no competing interests.

### Author Contributions

- Leanna P. Matthews conceived and designed the experiments, performed the experiments, analyzed the data, contributed reagents/materials/analysis tools, prepared figures and/or tables, authored or reviewed drafts of the paper, approved the final draft.
- Brittany Blades performed the experiments, contributed reagents/materials/analysis tools, authored or reviewed drafts of the paper, approved the final draft.
- Susan E. Parks conceived and designed the experiments, contributed reagents/materials/analysis tools, authored or reviewed drafts of the paper, approved the final draft.

### Animal Ethics

The following information was supplied relating to ethical approvals (i.e., approving body and any reference numbers):

The Institutional Animal Care and Use Committee (IACUC) at Syracuse University provided full approval for this research.

### Data Availability

Matthews, Leanna (2018): PlaybackDatabase_FigShare.xlsx. figshare. DOI 10.6084/m6.figshare.5129272.v1.

Matthews, Leanna (2018): R_FigShare.R. figshare. DOI 10.6084/m9.figshare.5129269.v1.

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
