# Peer review of "Female harbor seal (Phoca vitulina) behavioral response to playbacks of underwater male acoustic advertisement displays"

_PeerJ, doi:10.7717/peerj.4547_

## Round 0.1 · original submission · Major Revisions

I agree with both reviewers that you pose an intriguing question but may not yet have as definitive an answer as you'd like. If you can clarify your methods, fully state your assumptions, and limit your conclusions to what your data support, I think you will have an acceptable manuscript that advances your field and our understanding of harbor seals and their vocalizations.

Reviewer 1 ·

Basic reporting

advertisement displays”

The manuscript describes the response of captive female harbor seals to male harbor seal roars and control sounds. I like the question posed by the authors and the overall design of the study. The paper is well written and easy to follow. I am a bit concerned about the foundation on which the study rest and I also believe that there are other interpretations to the findings made by the authors and that as such the conclusions are an overreach. I have smaller comments regarding the methods.

Foundation
The theoretical framework began very promisingly in the first paragraph. However, the study relies on the results of an unpublished MSc thesis. [I realize that theses are peer-reviewed by faculty, yet I also realize that faculty have a stake on seeing theses finished, unlike anonymous reviewers of a manuscript.] I read the thesis by Nicholson (2000) and have serious concerns about the work. Relevant to this manuscript, I do not know how she determined that wild harbor seals were males and females, adults and sub-adults. It is not an easy task, sometimes outright impossible, and she did not describe at all how she determined age-sex classes. I am also not convinced that she was able to unambiguously ascertain whether a particular vocalization came from a specific male with just one recording hydrophone. My concerns about the thesis do not demerit the manuscript, but they do indicate that the theoretical foundation is not as strong as it appears in the manuscript. So my suggestion would be to present Nicholson’s findings as a hypothesis rather than as a fact. In addition, Nikolich et al. (2016) indicated that roars in their study site form a continuum, rather than a dichotomy of short versus long duration and high versus low frequency calls, so I think a brief discussion on the implications of these results to the framework in which this manuscript was placed and to its discussion is needed.

Experimental design

Methods
The authors posit that female harbor seals reacted more to male vocalizations than to control noise, something with which I agree; then they posit that roars are important in male-female vocalization. Although this could be the case, one could argue that females are reacting to male or even seal sounds given that the experimental design did not test for other types of vocalizations such as growls or the fact that roars can represent a continuum (see above). I realize that the design included unusual variations of the roars but it did not include other types of vocalizations and, as I mentioned before, the results from the second year indicated that only 50% of the sample size reacted according to the predictions of the hypothesis. Because females are in a captive environment, because of individual variation and context (something that the authors discussed for one of the subjects) and because of the limited sample size, I wonder how much the relatively ambiguous results (see above in interpretation) are a result of the novelty of a seal sound rather than the specifics of a roar. How would they have reacted if they heard a growl, for example? Given all these issues, I suggest that authors to tone down their conclusions.

The paper is very well written, but I was unable to figure out the experimental design regarding playbacks and trials (lines 165-168). It seems to me that more detail or a different way to provide the information is needed.

From where did the females come? Did they come from Elkhorn Slough? If not, what is the justification that they will react to roars from seals from another region? This is important given that seal sounds can vary at the individual and population level. How long were they in captivity? For how long were they exposed in the wild to harbor seal sounds? I believe more information is needed about the provenance of the females.

Less importantly, it is unclear to me if the measured levels of 149-156 dB were also applicable to the water controls (lines 182-183).

Validity of the findings

Interpretation
I believe the authors give strong conclusions to their results that are not warranted by the data. In year 2, the results of Pinky and Tater are very similar to those for year 1, so only 50% of the sample size did not approach manipulated roars and controls between the years, as predicted. Further, the differences in response between what are termed subordinate and dominant male roars were only one approach or contrary to predictions (60%). Given that the test was not significant, similar to the results of Hayes et al. (2004), I suggest that the authors provide alternative explanations. It is possible, like they suggest, that their predictions could still be correct; however, maybe the answer is not as straightforward as a high/short and low/long roar and it is more complicated, with the female response based on roar nuances and other characteristics. So, my suggestion is to present alternative interpretations and tone down the conclusiveness of the results.

Reviewer 2 ·

Basic reporting

The Methods, results, and discussion need to be clarified and re-organized.
Figure 1 is not helpful and needs axis labels and figure caption
Table 1 7 2 needs captions and I believe including which subjects were in estrous is needed.

Experimental design

Methods and analysis need much work/reorganization. They, themselves are fine, but the reporting of it needs improvement

Validity of the findings

My concern when answering this is that I am unclear based on some clarification needed through revised methods/analysis reporting.

Conclusion should go into more depth

Additional comments

The research question is interesting and the general work well done. The paper needs major revision in my opinion but I would like to see it published after revisions as something of this nature can add to the overall information in the field. Lab and field work complement each other - but this paper, as written, only scratches the surface or what I want to read and leaves me with more questions than answers in how the design was carried out.

-Please see all specific comments in the attached file.

Annotated reviews are not available for download in order to protect the identity of reviewers who chose to remain anonymous.

---

## Round 0.2 · Minor Revisions

Congratulations on a dramatically improved manuscript! You're very close to an acceptable paper, but I'd like to ask you to address at least the first comment of reviewer 1 (make tone and content of your abstract consistent with your conclusions). I'll let you decide how/whether to also alter your discussion as reviewer 1 has suggested.

Reviewer 1 ·

Basic reporting

The manuscript was well written and presented before, and it continues to be the case in this revised version.

Experimental design

I want to commend the authors for the revisions they made to their manuscript. The revised introduction makes the paper intriguing, with a framework that offers a more honest intellectual justification than the previous version.

Validity of the findings

The authors are upfront and candid about the limitations of their study, particularly the sample size. In this regard, the conclusions are for the most part excellent. However I believe two changes are important:

1- The abstract does not match the conclusions. The impression the reader has after reading the abstract changes after reading the paper, including the conclusions. I agree that females responded to male calls more than the control. Taking into account both years and all females, I am not quite convinced that there is preliminary evidence that females prefer LL male calls, I would say that results did not invalidate the hypothesis that females prefer LL male calls. I think it is up to the reader to make up her/his mind about that. Yet, the abstract presents the results on female approaches to LL calls first than the results of female responses to control and in stronger terms than in the conclusion. My suggestion is that the abstract follows the conclusion regarding the findings: female approaches to LL calls comes after female approaches to calls over control and qualify the statement on female approaches to LL calls.

2- I think the authors should either balance out or cut down the discussion (lines 292-328). The results did no support the hypothesis of female response to LL calls (both by looking at the figures and by running the statistical tests). They did not refute it either (as I said before). Yet the main thread in the discussion is about the hypothesis being correct or potentially correct and why that could be. The overall read I got from those lines was they were an overreach so at the very least they should be cut down. Personally, I would like to see a discussion of what if females actually do not respond to LL calls, what would that meand and why? It would certainly be speculative but so are lines 292-238 in the discussion.

---

## Round 0.3 · accepted · Accept

It is optional, but given that PeerJ is an open-access online journal, I'd like to encourage you to add to your submission a few recordings from your experiments (i.e. audio files in the web-browser-friendly .mp3 and/or .ogg formats). By providing at least a low frequency, long duration example, or ideally an example of each type of roar (LL, SL, SH, LH), your contribution would stand out in the literature. AFAIK, past publications have characterized the roars statistically, or in spectrograms, but have never been accompanied with recordings as supplementary information.

[# Staff Note - PeerJ can accommodate extensive supplemental files if you wish. Please liaise with the production staff if you wish to add any #]